# Implementing a Rural Natural Experiment: A Protocol for Evaluating the Impacts of Food Coops on Food Consumption, Resident’s Health and Community Vitality

**DOI:** 10.3390/mps5020033

**Published:** 2022-04-14

**Authors:** Éric Robitaille, Marie-Claude Paquette, Gabrielle Durette, Amélie Bergeron, Marianne Dubé, Mélanie Doyon, Geneviève Mercille, Marc Lemire, Ernest Lo

**Affiliations:** 1Institut National de Santé Publique du Québec, Montréal, QC H2P 1E2, Canada; marie-claude.paquette@inspq.qc.ca (M.-C.P.); gabrielle.durette@inspq.qc.ca (G.D.); amelie.bergeron@inspq.qc.ca (A.B.); marianne.dube@inspq.qc.ca (M.D.); marc.lemire@inspq.qc.ca (M.L.); ernest.lo@inspq.qc.ca (E.L.); 2Département de Médecine Sociale et Préventive, Université de Montréal, École de Santé Publique de l’Université de Montréal, Montréal, QC H3T 1A8, Canada; 3Centre de Recherche en Santé Publique, Université de Montréal et CIUSSS du Centre-Sud-de-l’Île-de-Montréal, Montréal, QC H3T 1A8, Canada; genevieve.mercille.1@umontreal.ca; 4Département de Nutrition, Université de Montréal, Montréal, QC H3T 1A8, Canada; 5Département de Géographie, Université du Québec à Montréal, Montréal, QC H3C 3P8, Canada; melanie.doyon@uqam.ca; 6Department of Epidemiology, Biostatistics and Occupational Health, McGill University, Montreal, QC H3A 1G1, Canada

**Keywords:** food environment, nutrition, interventions, accessibility, isolated

## Abstract

Background: Local food environments are recognized by experts as a determinant of healthy eating. Food cooperatives (coop) can promote the accessibility to healthier foods and thus improve the health of the population, particularly in remote rural communities. Objective: To measure the effects of implementing a food coop in a disadvantaged community with poor access to food. We have two main research questions: (1). Does the establishment of a food coop in rural areas described as food deserts have an impact on accessibility, frequency of use, food consumption, food quality, and ultimately the health of individuals? (2). Does the establishment of a food coop in rural areas described as food deserts have an impact on food security and community vitality? Design: A natural experiment with a mixed pre/post method will be used. The sample is composed of households that came from geographically isolated communities (population: 215 to 885 inhabitants) which qualified as food deserts and located in rural areas of Quebec (Canada). All communities plan to open a food coop (in the years 2022–2023), and as their opening will be staggered over time, participants from communities with a new food coop (intervention) will be compared to communities awaiting the opening of their food coop (control). Data collection was carried out at three time points: (1) before; (2) 1 to 5 months after; and (3) 13 to 17 months after the opening of the coop. Questionnaires were used to measure sociodemographic variables, dietary intake, residents’ health, and community vitality. Semi-structured interviews were conducted with community stakeholders. Results: Few natural experiments have been conducted regarding the impact of implementing food coops. Gathering concrete data on the effectiveness and processes surrounding these interventions through natural experiments will help to quantify their impact and guide knowledge users and policymakers to make more informed decisions.

## 1. Introduction

Unhealthy eating habits, being overweight and chronic disease jeopardize the health of Canadians and generate significant individual, social, and health service costs [1,2,3,4,5]. Preventing chronic diseases and overweight requires reducing energy intake and improving the food quality of the population [6]. This is especially true for people with low income who have been shown to have poorer diets [7,8]. 

Food security is defined as physical and economic access to sufficient, safe, and nutritious food that meets people’s energy needs and food preferences while enabling them to lead healthy and active lives [9]. For a community to be food secure, the following four conditions must be present: availability of food, physical and economic access to food, use of food, and stability in the presence of the three previous conditions [9]. Many rural communities in Quebec are food insecure because of limited or no access to food [10]. 

Healthy eating is determined by individual and physical, economic, political, and socio-cultural environmental characteristics [11,12,13]. Interventions to improve the population’s diet, therefore, require a portfolio of multi-targeted and multi-level strategies [14], including those aimed at improving the local food environment (LFE) [15,16,17]. The LFE is defined by the food supply (location and accessibility to several types of food stores) in municipalities and neighborhoods [18]. 

In both the United States and Canada, several cross-sectional studies have found a link between a lack of physical access to nutritious and affordable food, poor diet, and increased risk of obesity [19,20,21,22,23]. Public health organizations recognize the importance of developing or consolidating an LFE conducive to adopting and maintaining a healthy diet [24,25,26]. Promising interventions can be divided into the following four categories: (1) opening of conventional sources of supply (e.g., supermarkets) [27,28,29,30,31] or alternative sources of supply (e.g., solidarity grocery shops, public markets, and mobile markets) [32,33,34,35,36,37], (2) modifying the food offered inside stores (e.g., “healthy” convenience stores programs) [38,39], (3) land-use planning (e.g., zoning) [40] and (4) support for mobility (transport infrastructure) [41]. 

A limited number of studies have evaluated these interventions, either those related to the establishment of new sources of food supply regardless of whether they are conventional or alternative sources or in a rural and remote community. Positive effects on perceptions of access to healthy food, improvements in food quality, energy intake, as well as body weight, and some chronic diseases were documented following the implementation of supermarkets, particularly for the population living near the intervention site [29,30,37,42]. In Canada, opening a food coop in a food desert in Saskatoon has made it possible to reach households with lower incomes living nearby who bought more fruit and vegetables and fewer processed products than those living further away [43,44]. Research and evaluation around these initiatives is greatly needed [19,45,46,47], especially in rural areas that are typically characterized by a low population density and the presence of few food stores that propose a lower variety of foods that are of lesser quality and affordability [48]. 

These previous studies, however, are often of low quality due to their poor specifications of the methods and the nature of the intervention. Additionally, a majority of studies were carried out in the United States in an urban context [37,42,43]. The conclusions of these studies are hardly transferable to rural areas because of the type of interventions and the differences in LFE. In the wake of movements to transform food systems at the local level [49,50] and according to our community partners, the establishment of social economy enterprises such as food cooperatives would be a promising strategy to counter food deserts and improve the health of rural populations notably through community vitality [40,51,52,53]. Food cooperatives were targeted as most small rural areas do not have large enough populations to make a privately-owned food store economically viable.

The framework proposed by the Agency for Healthcare Research and Quality (AHRQ) Evidence-based Practice Center Program [54] guided the development of the research objectives for this project. A logic model (Appendix A) was developed to capture the diversity and complexity of the potential food and health effects of establishing a food cooperative. The logic model was constructed following the guide proposed by the Canadian National Collaborating Centre for Healthy Public Policy [55]. The main objective of this project is to assess changes in the local food environment on the health and vitality of the communities involved. We have two main research questions: (1). Does the establishment of a food coop in rural areas described as food deserts have an impact on accessibility, frequency of use, food consumption, food quality, and ultimately the health of individuals? (household level) (2). Does the establishment of a food coop in rural areas described as food deserts have an impact on food security and community vitality? (community level). Our first hypothesis is that the implementation of a food cooperative in areas qualified as food deserts will contribute to improving access to healthy food, the frequentation of this type of business will be greater, which will promote an increase in the consumption of healthy food, increase the quality of the food consumed and in the long term, will improve the health of the population. The second hypothesis is that the implementation of a food cooperative will contribute to increasing community food security and will help to increase the vitality of the communities.

We have fives sub-objectives: Objective 1:Evaluate the effects of the food cooperative on food consumption, residents’ health, and community vitality;Objective 2:Document the mobilization of community actors before and after the implementation of the food cooperative;Objective 3:Evaluate the effects of the intervention on food accessibility, use of the food cooperative, and food supply in the local environment;Objective 4:Analyze the socioeconomic and community contexts of the implementation of a food cooperative;Objective 5:Implement an integrated knowledge translation process (iKT) to improve practices throughout the project.

## 2. Experimental Design

A natural experiment research design with mixed sequential methods will be used. Qualitative data will be collected to facilitate interpretation and to contextualize the longitudinal quantitative data [56,57,58]. The natural experiment under study is the establishment of a food coop in rural remote communities. The opening dates, size, or type of foods sold in these coops are not under the control of the researchers [59]. The communities of Rivière-Saint-Jean and Magpie, Gallix and Rivière-Pentecôte located in the Côte-Nord (Québec) region were identified in collaboration with the Fédération des coopératives d’alimentation du Québec (Quebec’s food cooperative federation), a provincial organization supporting food cooperative projects [60]. The Côte-Nord region is a large region of the province of Québec bordered by the Saint-Laurence river which is sparsely populated and located 850 km north east of Montréal. The communities were selected as they are projected to open a food coop in 2021–2023, have low physical access to stores offering healthy food (supermarkets, grocery shops), and include disadvantaged socio-economics sectors (Figure 1). 

### 2.1. Rivière-Pentecôte

Rivière-Pentecôte has 317 households, or 885 inhabitants, according to the 2016 Canadian census [61]. The inhabitants of Rivière-Pentecôte live on average more than 6 km away from a grocery store. In the community, a convenience store-style food coop has been in operation for about 15 years. It offers staples, ready-to-eat meals, and fresh produce on-demand and to order. This coop is in the process of relocating due to its outdated location. The move is planned for 2022.

### 2.2. Gallix

This community consists of 304 households with a total of 674 inhabitants according to the 2016 census [61]. The inhabitants of Gallix live on average more than 19 km away from a grocery store. Gallix does not have a food service point. A project is underway to create and set up a food cooperative. The interim committee would like to see fresh produce and ready-to-eat meals sold at the cooperative. They also want to integrate a coffee corner as well as a gas and propane station.

### 2.3. Rivière Saint-Jean and Magpie 

In Rivière-Saint-Jean and Magpie, there are 112 households, or 215 inhabitants according to the 2016 census data [61]. The inhabitants of Rivière Saint-Jean and Magpie live on average more than 22 km away from a grocery store. A provisional committee is working on a project for a food cooperative that resembles a grocery outlet, and which will annex a gas station.

To grasp the complexity of implementing a food cooperative and its effects on food, participants, and the community’s health as adequately as possible, the project is divided into five activities (Figure 2): Prepare data collection tools;Interview key informants (n = 2–3/community) from the selected communities and other organizations involved in the implementation of food cooperatives (n = 3–4) before the implementation of the coops (T0) using semi-structured interviews. The key informants from the selected communities will be interviewed again in T1 or T2 to further understand the process of community mobilization;Collect information on the socio-economic characteristics and food environment of the selected communities;Collect data using Questionnaire surveys Visualisation, Evaluation, and Recording of Itineraries and Activity Spaces (VERITAS) [62,63] with quasi-experimental pre-post estimate [47] among 250 households with an adult 18 years old and older responsible for the food purchases of the household;Integrated knowledge translation process (iKT).

## 3. Procedure

### 3.1. Quantitative Data

#### 3.1.1. Study Population

The total targeted sample size is 105 households per community for a total of 315 households at T0. Households with an adult will qualify as potential members of the sampling frame. All communities will have collected data before any coop is opened (T0). Once the first coop is opened T1, data collection will begin in all communities (1 intervention group (IG), and 2 control groups (CG)). The communities that still do not have a coop at this point will serve as control groups. Participants will be questioned three times using the following intervals: (1) before the implementation of COOP (T0), (2) 1 to 5 months after the implementation of the cooperative (T1), and 13 to 17 months after the implementation of the cooperative (T2) [26,27,58] (Figure 1). 

#### 3.1.2. Sample Size Calculation

The multiple regression option of the Gpower software was used to determine the statistical power of the study and to ensure a sufficient sample size [64,65,66]. We specified a one-tailed t-test (equivalent to an F-test) on the coefficient of the exposure variable, 12 categories or levels distributed among the covariates, an alpha threshold of 0.95, and a statistical power level of 0.9. A sample size of 105 in the intervention community and in the control community (105 households) would be sufficient to detect a small effect size ($f^{2}$ = 0.06). This calculation considers an attrition rate (non-response and abandonment combined) of 50% at time T1 and does not depend on the choice of the exposure variable [67,68]. Considering only fruit and vegetable consumption and assuming a standard deviation of 1.78 for the difference in consumption, the sample size specified would correspond approximately to an effect size of 0.2 servings per day [30]. Our study, therefore, requires an almost exhaustive survey of the chosen communities. This should not be a problem because quasi-exhaustive sampling is often performed in the context of small rural communities [69]. 

#### 3.1.3. Recruitment and Retention of Participants

Recruitment in the context of quasi-experimental natural experiments has the following two main objectives: to obtain a representative sample of the target population and to recruit enough participants to meet the requirements of the sample size while ensuring a large enough sample is recruited to account for dropout rates. Our recruitment method is based on studies of similar populations [70] or with similar objectives [71]. Participants will be recruited via a Facebook campaign, polling firm, and local newspapers in which the person responsible for food purchases will subsequently fill out the questionnaire. Informed consent will be obtained at the beginning of the questionnaire after participants are made aware of the details of the research project (Appendix A). Recruitment and retention of study participants are essential in the case of a natural experiment with a quasi-experimental design [67] but can be challenging in rural communities [68,72]. Three recruitment and retention strategies that are known to be effective in rural areas will be used to maximize participant retention [68]:Community engagement: This will be facilitated by our iKT approach which includes communication with municipal and public health authorities and the region’s healthy living collaborative structures as research stakeholders.Increase awareness, knowledge, and understanding of the research: At the beginning of data collection (T0), a campaign to promote the research to citizens will be conducted through local and social media and at popular community locations (e.g., community centers, churches, post offices, gas stations). The objective is to recruit participants, explain the purpose of the research and the process, and the role of local and regional authorities in the project. This campaign will also enhance the profile of the project and research team [70].Social and financial supports for participants: Regular contact with participants through reminder letters and greeting cards will also be part of the retention strategies. Several studies have shown that financial incentives can improve the recruitment and retention of survey participants, particularly among socioeconomically disadvantaged populations. We, therefore, plan to use this strategy to facilitate recruitment and to increase the retention of participants [67,73,74]. The value of financial retribution to participants will be gradual and as follows: T0: $25, T1: $30, T2: $35. By adopting these strategies and based on the results of various studies, we foresee a retention rate of 70% at T1 and 50% at T2 [67,68].

The questionnaire will be completed online by the participating households. The questionnaire consists of 76 questions (Appendix A).

#### 3.1.4. Dependent, Independent, and Covariables

##### Primary Dependent Variable

The primary outcome variable is food consumption, assessed with the use of a fruit and vegetable intake questionnaire [75] and a food quality questionnaire [76] (Table 1). 

##### Secondary Dependent Variables

Secondary dependent variables include: Perceptions of food access from a set of nine questions each, measured on a five-point scale (strongly agree to strongly disagree) regarding quantity, variety, quality, price, accessibility of food near home or workplace [29,77,78,79];Community vitality and well-being will be measured using a combination of three measures of community vitality [80,81,82] and well-being. The final scale resulted in 19 questions on community resilience, citizen participation, community pride, and sustainable development. Questions will be answered on three or five-point Likert scales. A reliability analysis will be performed on the scale using lambda-6 [83];Frequency and location of shopping: The location of food shopping will be captured with the VERITAS platform. Additional questions on gardening and use of alternative resources (e.g., food pantries, market stalls) will complete the food frequency and location of shopping [84,85,86];Weight and height: one question on weight and one question on height will be used to calculate participants’ BMI. 

##### Independent Respondent Variables


Adult household food insecurity: The Household Food Security Survey Module (HFSSM) focuses on self-reports of uncertain, insufficient, or inadequate food access, availability, and utilization due to limited financial resources, and any compromised eating patterns and food consumption that may result. The HFSSM contains 18 questions about the food security situation in the household over the previous 12 months. Each question specifies a lack of money or the ability to afford food as the reason for the condition or behavior. The questions range in severity from worrying about running out of food, to children not eating for a whole day. Only questions that are specific to the experiences of adults in the household or the household in general (Adult Scale) will be used in this study [87].


##### Independent Community Variables

Concurrent with T0 data collection, assessments of the food environment and socioeconomic context will be conducted in the target communities. A database of food businesses from the Ministère de l’Agriculture, des Pêcheries et de l’Alimentation du Québec (MAPAQ) (Quebec’s ministry of agriculture) will be used and validated by our regional collaborators. This data will be supplemented by audits of community food businesses [88]. To characterize communities socioeconomically, data from the 2016 INSPQ Deprivation Index will be used [89]. The tool for auditing the community’s businesses, which was validated for supermarkets, must be adjusted for the context of smaller food stores [88]. The analyses will be repeated at T1 and T2, although in some instances no new data will be available at these data collection points as data are typically not collected every year. 

##### Covariables

Questions on covariables will be derived from the questions used in the census [90] or Canadian Community Health Survey (CCHS) [91] and will include age, housing tenure, marital status, gender, household income, education, number of children in the household, and modes of transportation used to purchase food.

#### 3.1.5. Data Protection and Management

According to the conservation rules in effect at the Université de Montréal, research documents and data must be retained for a minimum of 7 years after the end of the project. After this period, the Ethical Committee (Comité d’éthique de la recherche en santé (CÉRES)) recommends the de-identification of data, the destruction of identifying information, or any other measure which ensures long-term protection of the personal information collected. Retained data and their future uses are also subject to research ethics requirements.

### 3.2. Qualitative Data

Qualitative methods are ideal for gaining access to the experiences and perspectives of stakeholders [53], and thus for gathering a variety of views, realities, and issues. Key informants will be interviewed (semi-structured interview) before the opening of the cooperative and again following the opening to assess their views on the implementation process and to explore their perceived impact of the cooperative on the community. The key informants will be met with once for a semi-structured interview before the opening of the cooperative and then again following the opening of their cooperative, to assess their point of view on the implementation process and to explore their perceived impact of the cooperative on the community. Two members of the research team will be present during the interviews as one will guide the interview while the other will take notes. The interviews will last approximately one hour and will be conducted via videoconference or telephone.

#### 3.2.1. Recruitment of Key Informants

The provincial organizations who support food cooperative projects and helped us to identify the study communities also provided us with contact information of some members of the citizen committees in charge of the COOP project in each community. Semi-structured interviews will be conducted with these key informants. Other key informants will also be recruited by snowball effect, until data saturation. Interviews with members of provincial and regional organizations, who work with food cooperatives or the communities under study, will also be conducted. The key informant will provide written consent for audio and video recording under the assurance of confidentiality and anonymity. Financial retribution of 25$ will be provided after the interview. 

#### 3.2.2. Interview Guide

The interview process will follow the interview guide (Appendix A) and will be adapted to the interviewee and his or her role in the cooperative project. The construction of the interview guide is inspired by the work of Billion [92] on the role of trade and distribution actors in territorial food governance processes. We also introduced sections related to the local food security context to explore the role of alternative businesses on food security [93]. Additionally, several aspects of intersectoral collaboration (concertation in French) stated in Fortier’s reference framework [94] are also probed in the interview guide. Thus, it provides guidance for the key informants to discuss their role in the cooperative project, the needs that led to its development, the process and the steps to be carried out, the cooperative’s characteristics, community mobilization, and the required intersectoral collaboration among partners. The following topics will also be addressed: facilitating factors and barriers in implementing the cooperative, the cooperative’s impacts on the community and the citizens, and, finally, the measures put into place to ensure the sustainability of the cooperative. Open-ended questions will be asked so as not to direct the interviewer’s answers (e.g., In your opinion, what are the most important elements that explain why the cooperative project is progressing well?).

### 3.3. Analytical Strategies

#### 3.3.1. Quantitative Analysis

Differences between IG and CG will be estimated using intention-to-treat double-difference models [28] which is a “principle of comparing participants according to the treatment (coop) they were originally randomly assigned to, regardless of the treatment they received” [95] (Obj. 2 et 3). The double-difference model is a recognized method for assessing the impact of new food retail locations on diet and health with an IG and a CG [28,30,96,97,98]. First, comparability analyses between the two groups will be performed at T0. Means and standard deviations for continuous variables and percentages for categorical variables will be calculated. Subsequently, measures of mean differences will be calculated for each dependent variable between the values at T0, T1, and T2 for the IG and the CG. Estimates will be calculated from the double-difference model, which will be applied to measure the evolution between T0, T1, and T2 [30]:Y_ct_i_ = β_0_ + β_1c_ X_c_i_ + β_2_ X_t_i_ + β_3_ X_c_i_* X_t_i_ + ∑_j_ α_j_ Z_j_i_
(1)
where Y_ct_i_ represents the dependent variable corresponding to individual ‘i’ in community ‘c’ measured for period ‘t’ (=T0, T1 et T2), X_c_i_, X_t_i_ are indicator variables for community ‘c’ and period ‘t’, X_c_i_* X_t_i_ is the interaction term between the effect of community and period and Z_j_i_ represents the covariates. The coefficient β_3_ will show that the change between T1/T2 and T0 of the dependent variable is different between the IG and the CG, indicating an effect of the intervention [30]. The regression model will be weighted to account for sample attrition between T0, T1, and T2 and to ensure that the results are generalizable to the T0 sample [97]. The weights are equal to the inverse probability of response at T1 and will be estimated using logistic regression [97]. Multiple imputations will be used to perform the analyses with all participants who have been subject to data collection at T0, T1, and T2 [99,100]. All analyses will be performed using SAS version 9.4

#### 3.3.2. Qualitative Analysis

The interviews will be transcribed and then coded in the NVivo software (version 1.4.1) by two coders. An inductive analysis will be carried out, using a constant comparison method, to compare the new data to emerging theories [101]. Triangulation will be carried out using data collected via meetings with local partners and co-researchers [101].

### 3.4. Integrated Knowledge Translation Process (iKT)

The perspectives of various groups and sectors will influence our project activities. The way we work with our partners is guided by our iKT approach. This approach involves an ongoing dialogue between knowledge producers and knowledge users throughout the project. During project conception, high-level provincial knowledge users contributed actively and closely alongside us, notably in the framing of the project’s questions related to community mobilization, in the validation of recruitment and retention methods, and the quantitative and qualitative data collection tools. Additional regional and local knowledge users will be incorporated as the research unfolds. These partners will continue to participate in the interpretation, validation, dissemination, adoption, and appropriation of the knowledge acquired during the project. This back and forth communication between experts and knowledge users has already allowed and will continue to allow the project to be refined and improved as it progresses [102]. 

This approach is particularly well suited for a natural experiment in a real-world environment while promoting the appropriation of knowledge and its dissemination in other environments. 

The collaborative methods supported by the team take several forms that are as follows: (1). An advisory committee will be formed and composed of experts, knowledge users, and project staff. The committee will ensure that the project runs smoothly and will discuss elements of a strategic and scientific nature. It will advise the experts on the approach, methodological, ethical, and political aspects throughout the project. The members will read the project results, express their understanding and concerns, support the content and drafting of recommendations, contribute to the adaptation of the results and propose appropriate dissemination strategies to the different users. Three virtual meetings are planned throughout the project, as well as email and periodic newsletter exchanges. (2). A working group will also be established in the project area. The working group will be composed of a member of the project team (coordination) and knowledge users. The task of the working group will be to (i) monitor the progress of the project (e.g., recruitment and retention of participants) in the communities, (ii) provide its knowledge of the areas under study and its expertise in various fields (cooperatives, healthy lifestyle habits, municipal development, etc.) to promote optimal conditions for the project to be carried out, and (iii) ensure that the knowledge produced is useful to the communities and can contribute to the improvement of practices. The group will meet virtually three times during the project. 

Furthermore, iKT strategies and activities are planned throughout the project. These strategies will be validated with the advisory committee and the working group, and will include the following: (1) Brainstorming meetings (n = 4) consisting of interactive exchanges between researchers and knowledge users (representatives of the Inter-sectoral Tables, Public Health Departments, the community) will be organized at key moments of the project, i.e., before and after T0 and after T1 and T2, to react to the progress of the research and to learn about and comment on the results of the data collection. Community members will be identified through semi-structured interviews with key informants; (2) Communication of preliminary results through an internet platform, social networks, workshops, and documents, will be carried out to mobilize knowledge users. Our collaborations with the provincial coordination of the “Tables intersectorielles en saines habitudes de vie” will facilitate the iKT of our research project. Knowledge transfer strategies are also planned at the end of the project. These strategies will be validated with the advisory committee and the working group and include the following: (3) Publication of the research results in a collection that disseminates knowledge that can inform the choices of practitioners and decision makers involved in promoting healthy lifestyle habits and in scholarly journals; (4) A regional deliberative forum bringing together members of the steering committee, the working group, and members of the brainstorming meetings; (5) Publication of guides and tools aimed at helping different communities assess the impact and processes related to the implementation of a food business. 

## 4. Limits

The research project carries several risks, including that the implementation of the intervention and the community events are not under the control of the research team, which may explain the small number of studies that have evaluated this type of intervention to date. 

It is expected that recruitment in small rural communities will be difficult and therefore the statistical power could be insufficient to demonstrate the actual effects of opening food cooperatives on the variables of interest. This limitation, however, cannot be avoided as the purpose of the study is to specifically focus on small remote rural communities, and the pool of potential respondents thus remains small. 

Most of the data is self-reported. This represents a greater limitation for measures associated with diet, weight, and height as these are particularly prone to social desirability bias [103,104,105]. For example, the frequency of fruit and vegetable consumption could be over-reported.

The opinions and thoughts of respondents collected in the qualitative interviews will be specific to the individuals interviewed. Despite data saturation, it is likely that some perspectives or experiences will not be represented. Social desirability bias may also influence key informants to present their experiences more positively than to reflect reality. In addition to being non-generalizable, the analysis and interpretation of these data will be conducted by nutritionists with an interest in public health, which will color their interpretations of respondents’ experiences.

The study also has several strengths, including the use of a mixed-method design and the use of measures from tools validated in French. 

## 5. Expected Results

As mentioned, only a few studies have focused on assessing the effects of food retailing on health and nutrition, particularly in less urbanized and rural settings. We believe that the processes studied in our project will be transferable to other contexts. The iKT research project foresees the publication and dissemination of results and different tools concerning the methods for evaluating the implementation of a food business on food and health, on the barriers and facilitators of the implementation and on the interplay of stakeholders. 

The tools developed for data collection measures can be reinvested in providing guides that will help stakeholders and local communities. These data collection guides and their tools will help communities to: Draw up portraits of their community’s food environment (presence of food deserts, level of physical accessibility, food insecurity, community vitality);Take stock of socio-economic characteristics, food consumption, and food insecurity of the communities;Evaluate the effects of the implementation of their food cooperative;Illustrate, through examples from the communities, the mobilization process, the conditions for success, and the pitfalls to be anticipated when implementing a food cooperative.

The tools developed will provide a method by which to demonstrate the impact of coops and improve their activities. An evaluation of coop projects will be able to answer questions such as: Did you do what you set out to do? Should you bring in other partners? How are the partners working together? Are you reaching your target audience? Are you having the impact you intended? Were there any unexpected results? What lessons can you learn from this experience? In the long term, the results of the evaluation of our three communities will allow us to identify the factors that favor the implementation of strategies to consolidate the autonomy and food security of the communities involved as well as to improve access to outlets that promote healthy eating.

## Figures and Tables

**Figure 1 mps-05-00033-f001:**
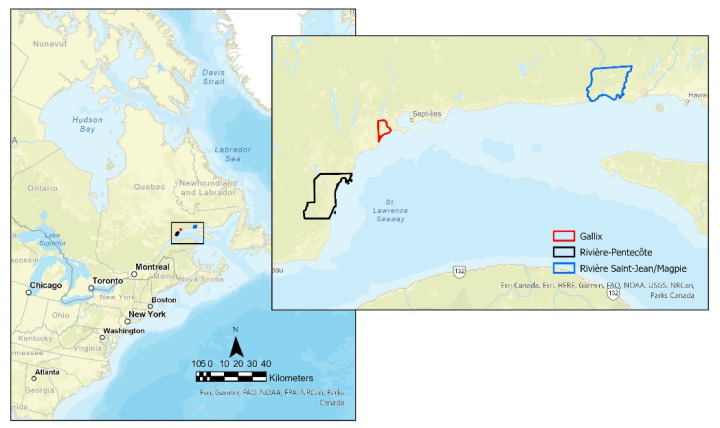
Localization of the communities.

**Figure 2 mps-05-00033-f002:**
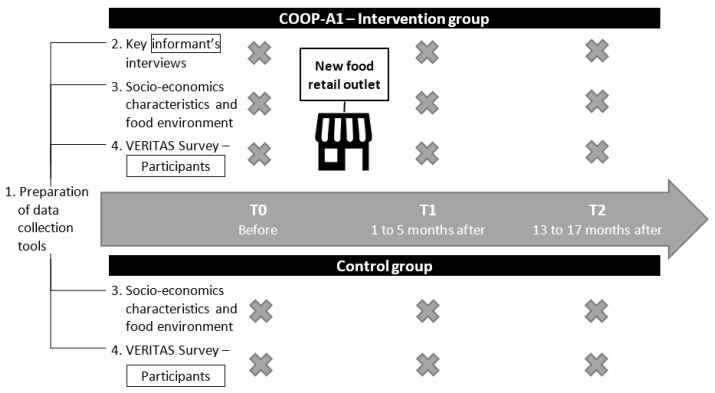
Research activities schema.

**Table 1 mps-05-00033-t001:** Quantitative study variables.

Variables	Methods	Measurement Tools (F = Validated in French)
Dependent variables		
Fruit and vegetable consumption	Surveys *	Fruit and vegetable module (CCHS) (F)
Food quality consumption	Surveys *	Brief food quality assessment tool (F)
Perception of the food environment	Surveys *	Nine questions each measured five-point scale (F)
Community vitality	Surveys *	Combination of three vitality and well-being scales
Body mass index (BMI)	Surveys *	Self-reported weight and size (F)
Independent variables		
Household food insecurity	Surveys *	Household Food Security Survey Module (F)
Attendance and food shopping locations	Surveys *	Interactive mapping tool to measure shopping locations.
Characteristics of the community’s food environment	GIS †	MAPAQ food business files (distance to nearest food retailer, distance to principal food retailer as identifier by participant, density of fast-food outlets)
Socioeconomic characteristics of communities	GIS †	INSPQ Deprivation Index and Vitality Index
Covariables		
Sociodemographic characteristics of individuals	Surveys *	Canadian Community Health Survey (CCHS) (F) and Census Canada (F), (age, income, education, marital and family status)

* Surveys will be completed at least 3 times: before the opening of the coop, 1 to 5 months after the opening of the coop, and 13 to 17 months after the opening of the coop. † It is possible that little to no new data will be available for the community food environment between data collection periods.

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
