# Peer review of "Implementing a Rural Natural Experiment: A Protocol for Evaluating the Impacts of Food Coops on Food Consumption, Resident’s Health and Community Vitality"

_mps, 2022, doi:10.3390/mps5020033_

Round 1
Reviewer 1 Report
The authors have outlined a protocol for a study evaluating the effects of Food Coops on consumer behavior and the community more broadly. While this area is of interest, the authors have not provided sufficient detail for a protocol. The authors should follow guidelines outlined for reporting of protocol papers, such as the SPIRIT 2013 Statement . While this statement is meant for clinical trial protocols, there are plenty of relevant domains that the authors should consult in revising the structure and content of their paper. The authors can consult the Journal’s Instructions for Authors, which highlight the need for “sufficient detail to allow others to replicate and build on published results.” The present manuscript does not allow for clear steps to replicate this process.
Additional recommendations are outlined below.
85 – I was not able to access the logic model (File S1 in Supplementary). Please provide the logic model and any other supplementary materials in your revision.
88 – Please describe the overarching goal of this project before outlining the objectives.
108 – 110 Please describe the selected communities in greater detail with referenced data to support these points.
191-192 Please provide much more detail about the measures used and how they relate to the target outcomes in this study. The measures and rationale for their use should be described in detail in the manuscript in English. Are these measures that have been validated in French?
What are the eligibility criteria for participants in these areas of study?
196 -197 Please provide a detailed description of the planned interviews with key informants. What questions will be asked? Are they based in theory? Who will be interviewing?
296 – As outlined in the Journal’s Instructions for Authors, the methods “should be described with sufficient detail to allow others to replicate and build on published results.” Please provide the interview guide and describe other methods in sufficient detail to allow for replication.
Author Response
"Please see the attachment."

Reviewer 2 Report
Dear Authors,
Thank you for this interesting paper (protocol) on evaluating the impacts of food coops on the food security of remote communities with limited access to food in Canada. The paper presents unique data. However, there are some issues to be improved:
1) The abstract should be more specific and include a research question and some data on the case study area, study population and study period (years? months?).
2) Keywords: "Rural" be replaced with another keyword (you already have it in the title).
3) Introduction:
(i) Numbered list should be starting with numbers on a new row.
(ii) The first paragraph is too long.
(iii) Please, include a research question before the objectives.
(iv) The theoretical background of food insecurity issues should be better explained.
4) Procedure:
(i) Study population needs more details of the participants.
(ii) Sample Size Calculation should be better referenced.
(iii) File S2 in Supplementary is in French. So, unfortunately, I can't evaluate it. If the paper is published in English it's better to have a translated version for review.
(iv) Specify what sociodemographic, dietary, health and community vitality indicators are measured in the Questionnaires.
5) The limitations of the study are not well-explained. Please, improve it.
6) Expected Results are poor-presented. Please, extend.
7) References should be formatted according to the style recommended by the Journal.
Good luck with your research.
Author Response
"Please see the attachment."

Round 2
Reviewer 1 Report
The authors have improved upon this manuscript. The newly added logic model is missing a lot of components of a logic model and appears to be just a simple flow chart. It is also unclear how these concepts relate to each other. Please check the spelling (access is spelled incorrectly) and add greater detail to the logic model. Please use a guideline for logic models, such as the one available at the link below.
Reviewer 2 Report
Dear Authors,
Thank you for revising the text.
Good luck with your research!
Author Response
Thank for your thoughtful comments.